# TLR2 and TLR9 Blockade Using Specific Intrabodies Inhibits Inflammation-Mediated Pancreatic Cancer Cell Growth

**DOI:** 10.3390/antib13010011

**Published:** 2024-02-01

**Authors:** Amrendra K. Ajay, Martin Gasser, Li-Li Hsiao, Thomas Böldicke, Ana Maria Waaga-Gasser

**Affiliations:** 1Division of Renal Medicine, Department of Medicine, Brigham and Women’s Hospital, Boston, MA 02115, USA; akajay@bwh.harvard.edu (A.K.A.); lhsiao@bwh.harvard.edu (L.-L.H.); 2Department of Medicine, Harvard Medical School, Boston, MA 02115, USA; mgasser@bwh.harvard.edu; 3Helmholtz Centre for Infection Research, 38124 Braunschweig, Germany

**Keywords:** TLR2 and TLR9, intrabodies, inflammation, pancreatic cancer, therapeutic intervention

## Abstract

Pancreatic cancer (pancreatic ductal adenocarcinoma, PDAC) remains a deadly cancer worldwide with a need for new therapeutic approaches. A dysregulation in the equilibrium between pro- and anti-inflammatory responses with a predominant immunosuppressive inflammatory reaction in advanced stage tumors seem to contribute to tumor growth and metastasis. The current therapies do not include strategies against pro-tumorigenic inflammation in cancer patients. We have shown that the upregulated cell surface expression of Toll-like Receptor (TLR) 2 and of TLR9 inside PDAC cells maintain chronic inflammatory responses, support chemotherapeutic resistance, and mediate tumor progression in human pancreatic cancer. We further demonstrated intracellular TLR2 and TLR9 targeting using specific intrabodies, which resulted in downregulated inflammatory signaling. In this study, we tested, for the first time, an intrabody-mediated TLR blockade in human TLR2- and TLR9-expressing pancreatic cancer cells for its effects on inflammatory signaling-mediated tumor growth. Newly designed anti-TLR2- and anti-TLR9-specific intrabodies inhibited PDAC growth. Co-expression analysis of the intrabodies and corresponding human TLRs showed efficient retention and accumulation of both intrabodies within the endoplasmic reticulum (ER), while co-immunoprecipitation studies indicated both intrabodies interacting with their cognate TLR antigen within the pancreatic cancer cells. Cancer cells with attenuated proliferation expressing accumulated TLR2 and TRL9 intrabodies demonstrated reduced STAT3 phosphorylation signaling, while apoptotic markers Caspases 3 and 8 were upregulated. To conclude, our results demonstrate the TLR2 and TLR9-specific intrabody-mediated signaling pathway inhibition of autoregulatory inflammation inside cancer cells and their proliferation, resulting in the suppression of pancreatic tumor cell growth. These findings underscore the potential of specific intrabody-mediated TLR inhibition in the ER relevant for tumor growth inhibition and open up a new therapeutic intervention strategy for the treatment of pancreatic cancer.

## 1. Introduction

Pancreatic ductal adenocarcinoma (PDAC) accounts for ~90% of pancreatic cancers and represents the fifth leading cause of death from cancer worldwide, with an overall survival rate of only 12% [1]. The current standard care involves surgical resection followed by chemotherapy [2]. Approximately 80% of PDAC patients are detected at the advanced or metastatic stage, where surgery is not an option and patients rely on palliative chemotherapy with nab-paclitaxel, gemcitabine, or mFOLFIRINOX [3]. This increases the lifespan for a mean of 10 months because of an early development of chemoresistance [2,4]. Due to the availability of targeted therapies by developing precision medicine, specific therapeutic regimens are gaining attention [5,6,7].

Pattern recognition receptors, specifically the Toll-like Receptor (TLR) family (TLR1–10 in mice; TLR1–9 and TLR11–13 in humans), are known receptors of ligands from various external pathogens (fungal, bacterial, and viral) and internal molecular particles and breakdown products (e.g., oxidized lipids, cell debris, genetic materials, and extracellular matrix) and execute an inflammatory response in the cells [8,9].

In addition to inflammatory signaling, TLRs have been shown to be involved in the proliferation and survival of epithelial and stromal cells, including fibroblasts and endothelial cells in multiple carcinoma [8,10,11,12]. As TLRs are expressed by immune cells and, to a lesser extent, by nonimmune cells, they are able to form heterodimers or bind to accessory molecules (e.g., CD14, CD36, and MD2), which leads to the activation of numerous downstream signaling cascades through Myeloid differentiation factor 88 (MyD88) [8,9,10].

TLRs have been shown to play important roles in tumor progression, as well as tumor inhibition, in pancreatic cancer [13]. For example, the activation of TLR7 and TLR9 in stromal cells or transformed epithelial cells can promote tumor progression [14,15]. On the other hand, immunomodulatory agonists for TLR2 and TLR9 can enhance tumor immunity [13,16,17,18]. Thus, TLR agonists can serve as a potential therapeutic target, as demonstrated through their topical application on certain skin tumors, inducing immune cell infiltration to the tumor site. Mechanistically, in chimeric and conditionally targeted PDAC mouse models, myeloid-specific TLR4 signaling executes the MyD88-independent TRIF pathway to promote tumor growth, whereas MyD88-dependent signaling inhibits tumor growth [19]. These in vivo model systems highlight the importance of the cellular context with respect to the anti- or pro-tumorigenic roles of TLRs in PDAC.

To understand the genes involved in PDAC progression, the whole transcriptome of patient tumor tissues, such as those from The Cancer Genome Atlas (TCGA), have identified gene signatures indicating innate immune responses and specific cell types, as well as cytokine signaling pathways, involved in established PDACs [20,21]. However, the surgical resection of tumors was from early stages I/II and thus excludes almost 80% of patients with the terminal disease (stages III/IV) [5,6,7,20,21].

TLR2 and TLR9 expression was demonstrated to be upregulated in human PDACs, which was confirmed by our previous data [22] and is listed in detail for human PDACs and established pancreatic cancer cell lines, including PANC-1 and BXPC-3 [23]. Thus, we utilized two human pancreatic cancer cell lines, PANC-1 (poorly differentiated advanced stage tumor, metastatic-derived) and BXPC-3 (moderately differentiated advanced stage tumor, non-metastatic). In summary, TLR3, TLR4, TLR7, and TLR9 induce inflammatory and immunosuppressive responses in immune cells, and TLR2, TLR4, TLR7, and TLR9 trigger a multitude of pro-tumorigenic activities in nonimmune cells [12]. To further analyze the potential of TLR-inhibiting strategies in overexpressing immune cells and nonimmune cells, with a focus on tumor cells, we developed intrabodies against the most interesting candidates, TLR2 and TLR9, and tested their inhibitory effects first in proof-of-principle experiments in inflammatory diseases [24,25]. More recently, we described in advanced stage pancreatic cancers overexpressed TLR2, TLR4, TLR7, and TLR9 predominantly in tumor cells rather than tumor-infiltrating immune cells, making TLR inhibition strategies in this tumor even more interesting. This concept became further supported through findings in xenograft tumor mouse models showing the potential of intrabodies to inhibit tumor growth of several cancers, such as gastric cancer [26,27,28].

In this study, we consequently targeted both TLRs (TLR2 and TLR9) using ER intrabodies in pancreatic cancer cells to prove their inhibitory effects on tumor growth. An intrabody is a recombinant antibody fragment designed to express inside the cells and directed to a specific target antigen present in various subcellular locations, including the plasma membrane, cytoplasm, nucleus, endoplasmic reticulum (ER), mitochondria, peroxisomes, and Golgi bodies. The localized targeting of the intrabodies is performed by adding a signal sequence for specific intracellular trafficking. KDEL is an ER retention peptide sequence on the C-terminal end of the intrabody. The KDEL sequence retains a protein from being secreted from the ER and facilitates its return to the ER [29,30,31,32]. Until now, intrabodies showed no off-target effect compared to RNAi and CRISPR-Cas. The advantages of intrabodies include very high specificity for the target, the possibility to knock down several protein isoforms by one intrabody, and the targeting of specific splice variants or even post-translational modifications [32]. The functional inhibition of TLR2 and TLR9 is mediated by the following mechanism: The translocation of TLR2 from the ER to the cell surface is inhibited, because the TLR2/ER intrabody complex is retained by the KDEL sequence [32]. As opposed to the surface expression of TLR2, intracellularly expressed TLRs such as TLR9 are transported from the ER to the endosomal compartment after stimulation with CpG DNA but are retained inside the ER resulting from the binding of the TLR9 ER intrabody to TLR9. Therefore, intracellularly expressed TLRs can only be targeted through specific intrabodies, not classical antibodies, reaching only extracellular targets. Here we found that these self-developed intrabodies specifically inhibit the proliferation of both advanced stage PANC-1 and BXPC-3 pancreatic cancer cells. 

## 2. Materials and Methods

### 2.1. Chemicals and Antibodies

#### 2.1.1. Chemicals

Phalloidin (Cat. no. R415) and OptiMEM were purchased from Thermo Fisher Scientific (Waltham, MA, USA). Brefeldin A solution 5 mg/mL (Cat no. 420601) was purchased from Biolegends (San Diego, CA, USA).

#### 2.1.2. Antibodies 

Anti-pSTAT3 Tyrosine 705 (Cat. no. 9145), anti-STAT3 (Cat. no. 9139), anti-Ki67 (Cat. no. 9449), anti-IL-6 (Cat. no. 12912), anti-Caspase 8 (Cat. no. 9746), Horseradish Peroxidase (HRP)-conjugated anti-rabbit IgG (Cat. no. 7074), and anti-mouse IgG (Cat. no. 7076) antibodies were purchased from Cell Signaling Technology (Danvers, MA, USA). Anti-TLR2 (Cat. no. ab213676), anti-TLR9 (Cat. no. ab37154), anti-Myc (Cat. no. ab32), anti-TNF-α (Cat. no. ab183218), and anti-Calreticulin (ab196159)-specific antibodies were purchased from Abcam (Danvers, MA, USA). GAPDH (Cat. no. G8795, Millipore Sigma, Burlington, MA, USA) was used as the loading control. Anti-rabbit, anti-rat, or anti-mouse Alexa488, Alexa 594, and Alexa 647-conjugated antibodies were purchased from Thermo Fisher Scientific (Thermo Fisher Scientific, Waltham, MA, USA). 

Two intrabodies towards human and murine TLR2 and TLR9 were generated from two antagonistic monoclonal antibodies (mAb) [24,25] to inhibit the function of both receptors by our laboratory.

### 2.2. Cell Lines

Pancreatic cancer cell lines BxPC-3 (Cat. no. CRL-1687) and PANC-1 (Cat. no. CRL-1469) were purchased from ATCC (Manassas, VA, USA). PANC-1 cells were grown in Dulbecco’s modified Eagle’s medium from ATCC (Cat. No. 30-2002) supplemented with 10% fetal bovine serum (FBS) (Gibco, Thermo Fisher Scientific, Waltham, MA, USA). BxPC3 wells were grown in RPMI-1640 medium (Cat. No. ATCC 30-2001) supplemented with 10% FBS. Both the cells were maintained at 37 °C in a humidified chamber supplemented with 5% CO_2_.

### 2.3. Methods

#### 2.3.1. Transfection

Cell transfection was performed using Lipofectamine 3000 reagent (Cat. No. L3000008 Thermo Fisher Scientific, Waltham, MA, USA) per the manufacturer’s protocol. Briefly, BxPC-3 or PANC-1 cells were plated in a six-well plate and allowed to adhere overnight in the complete medium in a humidified CO_2_ incubator at 37 °C. The medium was replaced with OptiMEM, and 1 μg pCMV Myc, pCMV Myc NCAM1, pCMV Myc TLR2, or pCMV Myc TLR9 plasmid was mixed with 20 μL P3000 in 500 μL OptiMEM and incubated for 5 min at room temperature. Thirty microliters of Lipofectamine 3000 were diluted in five hundred microliters of OptiMEM and incubated at room temperature for 5 min. The plasmid and Lipofectamine 3000 mixture was combined and incubated at room temperature for 20 min before adding it to the cells. The cells were incubated at 37 °C for 6 h and washed thrice. The cells were then incubated in a complete medium and used for the endpoint analyses at the indicated time points. 

#### 2.3.2. Generation of Stable Cells

For the development of stable cells expressing Myc protein or intrabodies against NCAM1, TLR2, and TLR9, transfected cells were selected against G418. A dose of 100 μg/mL and 400 μg/mL of G418 was used based on our dose–response curve for BxPC-3 and PANC-1 cells, respectively. After transfecting the cells with 1 μg of plasmids using Lipofectamine 3000, as mentioned above, the cells were washed after 6 h. The cells were washed three times with a complete medium and incubated for another 18 h with a complete medium. The cells were then incubated with G418 in the complete medium for 3 days, and the medium with the same concentration of G418 was replaced every 3 days. After 6 weeks, selected cells that were growing were pooled together. The intrabody expression was confirmed by immunofluorescence.

#### 2.3.3. Cell Proliferation Assay 

Cell proliferation was assayed by the CellTiter 96 AQueous One Solution Cell Proliferation Assay, as suggested by the manufacturer’s protocol (Promega Cat. no. G3580, Promega, Madison, WI, USA). Briefly, cells were plated in a 96-well plate with five replicates per condition. The cells were transfected with pCMV Myc, pCMV Myc NCAM1, pCMV Myc TLR2, or pCMV Myc TLR9 plasmids using Lipofectamine 3000. Five hundred nanoliters of P3000 reagent were mixed with 0.2 μg of each plasmid in 25 μL of OptiMEM to transfect each well. In another tube, five hundred nanoliters of Lipofectamine 3000 were mixed with OptiMEM and incubated for 5 min at room temperature. The plasmid and Lipofectamine 300 mixtures were combined and incubated for 20 min at room temperature. The DNA–Lipofectamine 300 complex was then added to the cells dropwise, and the cells were washed six hours post-transfection. The cells were then incubated with a complete medium for 72, 120, and 168 h. Forty microliters of CellTitre one solution were added to 200 μL medium 2 h before each time point. Plates were read using a spectrophotometer (Molecular Devices, San Jose, CA, USA) at a wavelength of 490 nm. pCMV Myc transfected cells were considered as 100% proliferation, and graphs were plotted with respect to the Myc transfected cells.

#### 2.3.4. Live/Dead Cell Staining Using Propidium Iodide (PI)

To evaluate the amount of live and dead cells after TLR2 and TLR9 intrabodies expression, the cells were transfected with pCMV Myc, pCMV NCAM1, pCMV TLR2, and pCMV TLR9 plasmids as described above and incubated for 48 h. The cells were then harvested by trypsinization and incubated with 100 µg/mL PI solution in PBS for 30 min. The cells were then acquired on a Cytek Aurora flow cytometer and analyzed using FloJo software, (v9, Ashland, OR, USA). The percentage of PI-positive cells was plotted as dead cells.

#### 2.3.5. Brefeldin A Treatment

To treat the cells, Brefeldin A, stock solution was diluted to 5 μg/mL in the complete medium, and the cells were treated for 12 h. The cells were then fixed and processed for immunofluorescence staining.

#### 2.3.6. Immunofluorescence

Immunofluorescence staining on the cells was performed as previously described [33,34,35]. Briefly, the cells were washed with PBS and fixed with 4% paraformaldehyde (PFA) for 15 min at room temperature. Samples were permeabilized using 1% Triton X-100 for 15 min for nuclear and cytoplasmic staining at room temperature or left unpermeabilized for the membrane protein. The cells were washed twice with PBS and blocked with two percent normal sheep serum containing 5% BSA for 1 h at room temperature. The primary antibodies (1:200 dilutions) were incubated overnight at 4 °C in five times diluted blocking buffer containing 0.1% Tween 20. After three washes of PBST (0.1% Tween 20), fluorescence-labeled species-specific secondary antibodies (Thermo Fisher Scientific) were added and incubated for 1 h at room temperature. The slides were washed thrice with PBST and mounted with a medium containing DAPI (Vector laboratories). Three to five images were captured using a 60× objective on a Nikon Confocal Imaging system (Nikon C1 Eclipse, Nikon, Melville, NY, USA).

#### 2.3.7. Quantitation of Immunostaining

Calculation of the Mean Fluorescence Intensity (MFI): ImageJ (version 1.54) was used for image analysis and its quantitation. The mean fluorescence intensity of the respective channels was calculated after setting up a threshold for all the images. The intensity was normalized to the total number of cells by counting the DAPI-positive cells. The fold change was plotted with respect to the control cells. For counting the number of positive nuclear staining, the cells were counted for their respective channels and normalized to the number of cells by counting the nuclei. The fold change was plotted with respect to the control cells.

#### 2.3.8. Counting of Ki67-Positive Cells and Plotting the Graphs

For counting the number of Ki67-positive cells, ImageJ was used, and after setting up a threshold, the total number of KI67-positive cells was calculated. The total number of DAPI-positive cells was counted using ImageJ (version 1.54). The ratio of Ki67-positive cells over DAPI-positive cells was calculated. The ratio of cells was plotted as the fold change with respect to the pCMV Myc NCAM1 as the control.

#### 2.3.9. Colocalization Analysis

For the colocalization studies, we used ImageJ. After selecting the channels, the colocalization coefficients were measured and plotted as arbitrary units.

#### 2.3.10. Phase Contrast Imaging

The cells were fixed with 4% PFA and imaged using a phase contrast microscope. The imaging was performed using an Echo Revolve microscope (Echo, CA, USA) using objectives of 4× or 10× magnification.

#### 2.3.11. Western Blotting

Western blotting was performed as previously described [34]. Briefly, the cells were pelleted after the endpoints and were incubated with RIPA buffer containing protease and phosphatase inhibitors for 2 h at room temperature. The cells were vortexed every 30 min and sonicated to break the DNA using a sonicator. Cell lysates were cleared using centrifugation at 15,000 rpm for 30 min at 4 °C. An equal amount of protein (30 μg) was loaded onto the precast gradient (4–20%) polyacrylamide gel. The gel was resolved at 100 V for 1 h in a running buffer (25 mM Tris, 192 mM Glycine, and 0.1% SDS, pH 8.3). Following gel resolution, the gel was washed with deionized water, and the PVDF membrane was preconditioned in 100% methanol for 1 min, followed by soaking in a transfer buffer (25 mM Tris and 192 mM Glycine, with 20% methanol, pH 8.3). The protein was transferred onto the PVDF membrane, and blocking was performed with 5% nonfat milk in TBST for 1 h at room temperature. The membranes were washed and probed with 1:1000 dilutions of TLR2 (Cat. no. ab213676, Abcam, Waltham, MA, USA), TLR9 (Cat. no. ab37154, Abcam), pSTAT3 Y705 (Cat. no. 9145S, Cell Signaling Technology, Danvers, MA, USA), STAT3 (Cat. no. 9139S, Cell Signaling Technology), and Myc (Cat. no. ab32, Abcam)-specific antibodies. GAPDH (1:5000 dilutions) (Cat. no. G8795, Millipore Sigma, Burlington, MA, USA) was used as the loading control. The membranes were washed thrice for 10 min each with PBST at room temperature. Species-specific HRP-conjugated secondary antibodies (1:5000) were incubated in 5% nonfat milk for 1 h. The membranes were washed thrice for 10 min each with PBST at room temperature. Blots were developed using Luminata Forte HRP (Millipore Sigma) reagent, and images were captured using the Syngene imaging system (Fredrick, MD, USA).

#### 2.3.12. RNA Isolation, cDNA Synthesis, and Quantitative Real-Time PCR

Total RNA was isolated from cell pellets using the RNeasy Micro or Mini Kit (Qiagen, Germantown, MD, USA). Briefly, the culture medium was removed from the cells, and the cells were lysed by adding and pipetting up and down 10 times with 350 μL of RLT buffer. Cell lysates were collected with a rubber policeman and collected into a microcentrifuge tube (not supplied). The lysates were passed at least 5 times through a blunt 20-gauge needle. A total of 350 μL of 70% ethanol was added and mixed by pipetting. Seven hundred microliters of each sample were transferred into the RNeasy spin column on the vacuum manifold. After the samples were passed through 700 µL of buffer, RW1 was added to each RNeasy spin column. Next, 500 µL of buffer RPE was added to each RNeasy spin column and repeated twice. The RNeasy spin columns were removed from the vacuum manifold and placed in 1.5 mL collection tubes. Fifty microliters of RNase-free water was added to each spin column and centrifuged at 10,000 rpm for 1 min at room temperature. Quantitation of the RNA was performed using a nanodrop (Thermo Fisher Scientific). First-strand cDNA synthesis was performed for each RNA sample from 0.5 μg of total RNA using the SuperScript reverse transcription kit (Cat. No: 11904-018, Thermo Fisher Scientific, Waltham, MA, USA). Preparation of the RNA/primer mixture in a sterile 0.5 mL tube was as follows
**Component****Amount**RNAx µL10 mM dNTP1 µLRandom hexamers (50 ng/µL)2 µLDEPC waterto make 10 µL

The RNA/primer mixture was incubated at 65 °C for 5 min and immediately transferred on ice for at least 1 min. The reaction mix (2×) was prepared in a separate tube by adding each component in the indicated order.
**Component****Amount**10× RT buffer2 µL25 mM MgCl_2_4 µL0.1 M DTT2 µLRNaseOUT™1 µL

Nine microliters of the 2× reaction mix was added to each RNA/primer mixture, mixed gently, and collected by brief centrifugation. The samples were incubated at room temperature (~25 °C) for 2 min. One microliter of SuperScript™ II RT was added to each tube and incubated for 10 min at room temperature. First-strand synthesis was carried out at 42 °C for 50 min. The reaction was terminated at 70 °C for 15 min and chilled on ice. One microliter of RNase H was added to each tube and incubated for 20 min at 37 °C before PCR.

#### 2.3.13. TaqMan RT-PCR

RNA from the cells was isolated using the miRNeasy Mini Kit (Qiagen, Germantown, MD, USA), as mentioned above. TaqMan PCR was performed as previously described [36]. Two microliters of five-times diluted cDNA were used for PCR using the TaqMan™ Universal PCR Master Mix with FAM-labeled gene-specific probes (Thermo Fisher Scientific) in duplicate on a QuantStudio 7 thermal cycler (Thermo Fisher Scientific). 

The PCR reaction was carried out as follows:
**Component****Amount**TaqMan™ Fast Advanced Master Mix (2×)5 µLTaqMan™ Assay (20×)0.5 µLNuclease-free water2.5 µLcDNA2 µL

The reaction was held at 95 °C for 20 s and with 40 cycles of denaturation at 95 °C for 1 s and annealing/extension at 62 °C for 20 s. The fold change was calculated by subtracting the Ct values of the candidate gene from the reference gene (delta Ct method). GAPDH served as the reference gene, and the fold change was presented with respect to experimental controls. A list of the TaqMan assays is shown in Table 1.

#### 2.3.14. Immunoprecipitation

Immunoprecipitation (IP) was performed as previously described [34]. Briefly, PANC-1 cells were plated in a 10 cm culture dish at a density of 70% and incubated in the complete medium overnight. The next day, the cells were transfected with pCMV Myc NCAM1, pCMV Myc TLR2, or pCMV Myc TLR9 plasmids using Lipofectamine 3000. One hundred and twenty microliters of P3000 and 10 μg plasmid were added to 1 mL of OptiMEM and were incubated for 5 min at room temperature. One hundred and twenty microliters of Lipofectamine 3000 were incubated with 1 mL of OptiMEM and incubated for 5 min at room temperature. The plasmid and Lipofectamine solutions were combined and incubated for 20 min at room temperature. The complex was then added to the cells dropwise. The cells were washed six hours post-transfection and incubated with a complete medium for 48 h. The cells were washed with PBS and lysed in 500 μL immunoprecipitation (IP) buffer (20 mM Tris HCl, pH 8.0, 137 mM NaCl, 10% glycerol, 1% NP-40, and 2 mM EDTA) containing a protease inhibitor cocktail. Lysis was performed on ice by incubating the samples for 1 h by gently swirling them every 15 min. The lysate was cleared by centrifuging at 15,000 rpm for 30 min at 4 °C. Fifteen hundred micrograms of protein were incubated overnight at 4 °C with 4 μg goat polyclonal anti-Myc antibody on a rocker. Fifty microliters of protein A/G agarose were added and incubated for 2 h at room temperature. The beads were washed three times with IP buffer. The immune complex was eluted by adding 1× SDS loading dye and heating at 100 °C for 10 min, and immunoblotting was performed to detect intrabody myc, TLR2, or TLR9 proteins, as discussed in the Western blotting section.

### 2.4. Statistical Analyses

Statistical analyses were performed using a two-tailed unpaired *t*-test to compare two groups with a minimum of three samples each and post hoc Bonferroni testing. Data are presented as the mean ± SD of three individual experiments A *p*-value of less than or equal to 0.05 was considered statistically significant. Data are presented as the mean ± SD. GraphPad Prism 9.0 (GraphPad Software, Inc., La Jolla, CA, USA) was used for all statistical analyses.

## 3. Results

### 3.1. TLR2 and TLR9 Intrabodies Cause Pancreatic Cancer Cell Death

To investigate the potential of TLR2 and TLR9 inhibition in pancreatic cancer cells and its efficacy in tumor cell death, we developed pCMV Myc NCAM1, pCMV Myc TLR2, and pCMV Myc TLR9 plasmids expressing NCAM1 intrabodies (control intrabodies) and TLR2- and TLR9-specific intrabodies. First, we tested the transfection efficiencies of both BxPC-3 and PANC-1 tumor cells by transfecting them with pCDNA3 GFP plasmid. This demonstrated, for poorly differentiated PANC-1 cells, an increased transfection efficiency (60%) compared to moderately differentiated BxPC-3 cancer cells (35%) (Appendix A–C), suggesting a dependency on different growth characteristics and/or growth dynamics. We found that TLR2 and TLR9 intrabodies resulted in cell death as compared to the controls (pCMV Myc and pCMV Myc NCAM1) of BxPC-3 and PANC-1 cells (as shown by phase contrast imaging at 4× magnification, Appendix A) and 10× magnification (Appendix A). Next, we performed MTS assays to quantitate the cell proliferation in a time-dependent manner. Our results showed a significant decrease in cell proliferation by TLR2 and TLR9 intrabodies at 72, 120, and 168 h post-transfection as compared to the control plasmids (Figure 1A,B). Next, we performed propidium iodide (PI) staining for detecting dead cells using FACS analysis, and we found a significant increase in cell death in BxPC-3 (~30%) and PANC-1 (~30%) cells expressing TLR2 and TLR9 intrabodies as compared to the control cells (Figure 1C,D). Further, we investigated the proliferation marker Ki67, and consistent with our previous data, we found a significant decrease in Ki67-positive cells in TLR2 and TLR9 intrabody-expressing cells (Figure 1E). Thus, our results confirm that TLR2- and TLR9-specific intrabodies inhibit pancreatic tumor cell growth.

### 3.2. TLR2 and TLR9 Intrabody-Mediated Pancreatic Cancer Cell Death through Inhibition of the STAT3 Phosphorylation Pathway

To detect signaling pathways that may be involved in pancreatic cancer cell death through TLR2 and TLR9 intrabody-mediated intracellular inhibition, we performed a Western blot analysis of STAT3 phosphorylation changes in the treated cells. Our results demonstrated a significant decrease in phospho-STAT3 and an increase of cleaved Caspase 3 by TLR2 and TLR9 intrabodies as compared to the control cells in both BxPC-3 and PANC-1 cells (Figure 2A,B). Further, in our quest to identify the executor pathway for pancreatic cancer cell death, we found Caspase 8 to be significantly increased in cells expressing TLR2- and TRL9-specific intrabodies compared to the control cells (Figure 2C). Thus, our data indicate that TLR2- and TLR9-mediated pancreatic cancer cell death occurs through pSTAT3 signaling, which is, finally, executed through Caspase 8.

### 3.3. Inhibition of TLR2 and TLR9 by Its Specific Intrabodies Blocks Inflammatory Mediators IL-6 and TNF-α

Downstream pathways that become upregulated by TLR2 and TLR9 result in increased inflammatory signaling [37,38], which was suggested to support cancer progression over time [39,40]. Thus, we asked whether TLR2 and TLR9 intrabodies reduce the inflammatory mediators IL-6 and TNF-α. Our results show a significant decrease in *IL-6* mRNA in BxPC-3 (Figure 3A) and PANC-1 cells (Figure 3B). This was further confirmed by a decrease in IL-6 protein, as shown by immunostaining PANC-1 cells (Figure 3C). In addition to IL-6, our results also show a significant decrease in *TNF-α* mRNA in BxPC-3 (Figure 3D) and PANC-1 cells (Figure 3E), which was further confirmed by a decrease in TNF-α protein expression in PANC-1 cells (Figure 3F).

### 3.4. TLR2 and TLR9 Intrabodies Bind to TLR2 and TLR9 Proteins to Retain These Proteins in the Endoplasmic Reticulum

To evaluate the binding of TLR2 and TLR9 intrabodies to their respective target proteins and their ER retention, we performed immunostaining for the intrabodies, target protein, and Calnexin (ER marker). We found colocalization of these three proteins, indicating their binding (Figure 4A–C). Quantitation of the colocalization of Myc with their respective antigens (TLR2 and TLR9) showed at least 50% colocalization (Figure 4D) compared to NCAM1. Next, to confirm the binding of the TLR2 and TLR9 intrabodies to their target proteins, we employed an immunoprecipitation assay to pull down the bound complex using an the anti-Myc antibody, and our results showed a significant binding of TLR2 or TLR9 protein with intrabodies as compared to IgG (Figure 4E,F). Thus, these results show the binding of TLR2 and TLR9 intrabodies to their respective protein with a significantly inhibiting TLR function by their ER retention.

## 4. Discussion

PDAC has a recurrence rate of 85% after surgical resection, and the 5-year survival even after complete surgical resection is still less than 30%, leading to high mortality among solid carcinomas. Significant improvements have been made in surgical techniques and postoperative results, but the overall survival rate has not yet improved even above 10% [41,42]. The heterogeneous microenvironment is characterized through genetic and epigenetic modifications that cause molecular changes; favor tumor growth; and, finally, lead to chemoresistance, resistance to radiotherapy, and reduced overall survival [43,44,45]. This challenge demands novel molecular targets and the development of therapeutic strategies for the better survival of patients with such a devastating tumor disease.

Recent advances in tumor immune therapy show the promise to treat several types of cancer based on their specifically expressed molecular receptors [46,47,48]. In addition, the combination therapy of chemotherapeutic agents or inhibition of overexpressed genes by siRNA and CRISPR/Cas9 molecules offer growth inhibition and specific targeting [13,49], although this type of gene therapy has its clinical limitations and has not yet been introduced into routine practice. Antibody-based cancer therapies, meanwhile, have become a cornerstone in tumor immune therapy, showing beneficial results in increasing the lifespan of patients with different types of cancer [50]. Nevertheless, traditional antibodies have the limitation of targeting only cell surface-expressed antigens but not receptors inside the tumor cell. Moreover, antibodies that clearly work in advanced colon cancer through a signaling blockade during angiogenesis did not show relevant benefits in patients with pancreatic cancer. To overcome the limitation of desirable intracellular signaling blockades in tumor cells, intrabodies have been designed that are capable of targeting intracellular tumor-associated antigens and neoantigens in cells and xenograft mouse models [27,29,51,52,53,54,55,56,57,58,59,60]. This strongly supports the further development of relevant intrabodies, with the goal of setting up clinical trials.

Until now, intrabodies have demonstrated no off-target effects in contrast to siRNA [61,62] and CRISPR/Cas9 techniques [63]. We and others have shown that TLR2, TLR4, and TLR9 expression is significantly increased in PDAC [22,64,65]. Targeting TLR-mediated inflammatory signaling and, particularly, TLR2 and TLR9 may therefore be beneficial for patients with pancreatic cancer. We performed proof-of-concept studies to investigate whether TLR2 and TLR9 intrabodies that we generated [24,25] could inhibit pancreatic cancer cell growth, decrease inflammatory markers, and induce apoptosis. Our results showed significantly inhibited pancreatic tumor cell growth by TLR2 and TLR9 intrabodies, significantly decreased inflammatory cytokines IL-6 and TNF-α, and increased apoptotic mediators Caspase 8 and cleaved Caspase 3. Specifically, Goumas et al. found that pancreatic cancer cells (BxPX-3 and PANC-1, among others) showed increased IL-6 expression [66]. In addition, patients with pancreatic cancer showed increased levels of plasma IL-6 protein [67]. The study confirms that TNF-α expression is increased in PDAC compared to normal cells, and its inhibition reduces tumors [68]. Thus, IL-6 and TNF-α are being investigated in clinical trials for pancreatic and other cancers. Our results are consistent with findings from other groups showing that blocking TLR4 or TLR9 potentiates pancreatic cancer cells to apoptosis [69,70,71]. Further, at the molecular level, we found that, after the intrabody-mediated blocking of TLR2 and TLR9, these two proteins were accumulated in the endoplasmic reticulum (ER). Due to ER accumulation, the functional effective concentration of TLR2 and TLR9 proteins was decreased. Moreover, we found that TLR2 and TLR9 intrabodies bind to their specific antigens. Our colocalization quantitation analysis showed at least a 50% colocalization of TLR2 and TLR9 with Myc-intrabodies, which is significantly high, because Myc intrabody expression was found only in 50% of the cells following transfection. A recent study confirmed that TLR2 knockout in mice showed a reduced tumor growth of colon cancer, and TLR2 downregulation in cancer cells reduces its proliferation in colorectal cell lines HCT116 and HT29 [72]. We found that TLR2 and TLR9, along with TLR4, play an important role in the growth of pancreatic cancer [22]. In addition, TLR9 inhibition has been shown to inhibit tumor growth and apoptosis in neuroblastoma [73]. In esophageal cancer, TLR9 expression is associated with aggressiveness and tumor grades [74]. TLR9 in pancreatic stellate cells has been shown to increase the proliferation of pancreatic cancer cells via paracrine signaling [15]. Thus, these reports support our hypothesis that the inhibition of TLR2 and TLR9 reduces cancer cell growth. Therefore, based on previous studies performed by other investigators to inhibit TLR4 and TLR9 using other strategies [64,65,75] and our published data on the inhibition of the sialyation of polysialyltransferases ST8SiaII and ST8SiaIV in rhabdomyosarcoma tumor cells [59], we hypothesize that these intrabodies may show promise in reducing tumor growth in xenograft mouse models.

Tumor-specific immunotherapy with antibodies is limited by the availability of tumor-specific neoantigens, which are expressed only on pancreatic tumor cells. Most antibodies against pancreatic cancer target tumor-associated antigens, which are overexpressed on tumor cells compared to normal cells [76]. We cannot exclude that, if we use an antitumor-associated antigen antibody fused to a lipid nanoparticle, the intrabody mRNA can be delivered to normal cells as well. This could be circumvented if we would use a TCR-like antibody recognizing a neopeptide/MHC I complex on the tumor cells. The new strategy can be used for other types of cancer, e.g., oral squamous cell carcinoma expressing TLR9 [77] and gastric cancer expressing TLR4 [78,79].

Our findings are highly relevant in pancreatic cancer therapy, particularly for TLR9, which cannot be targeted with classical antibodies due to its intracellular expression. Furthermore, the possibility of specifically targeting pancreatic tumor cells through the delivery of TLR intrabody DNA with pancreatic tumor-specific, capsid-modified, adenovirus-associated vectors or TLR intrabody mRNA embedded in pancreatic tumor-specific lipid nanoparticles [29] will favor clinical applications of the intrabodies.

Generally, pancreatic tumor-specific delivery is performed with antibody–drug conjugates and aptamer–drug conjugates. The targets are predominantly tumor-associated antigens. SHR-A1403ADC is an antibody–drug conjugate composed of an antibody against c-Met (C-mesenchymal epithelial transformation factor, a hepatocyte growth factor receptor) and a toxin-inhibiting proliferation, migration, and invasion of tumor cells [80]. In addition, aptamers (single-stranded oligonucleotides) can target cytotoxic drugs to pancreatic tumor cells. APTA-12 is gemcitabine fused to antibody AS1411 binding to nucleolar proteins that are highly expressed in tumor cells. Gemcitabine (2′,2′-difluorodeoxycytidine) is a nucleoside analog of deoxycytidine [81]. Furthermore, different nanoparticles have been described specifically targeting toxins, DNA, and shRNA to pancreatic tumor cells [82,83,84]. In summary, our intrabodies show the potential for a novel and specific treatment strategy for patients with pancreatic cancer.

## Figures and Tables

**Figure 1 antibodies-13-00011-f001:**
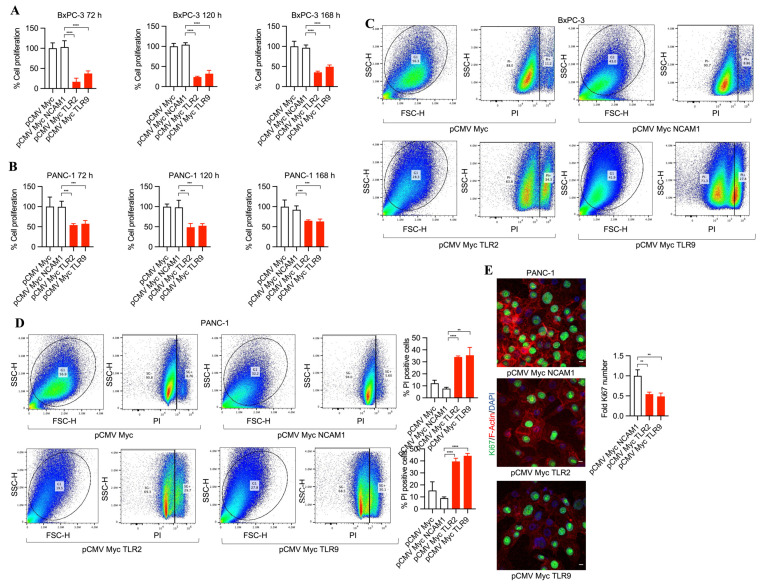
TLR2 and TLR9 intrabodies cause pancreatic cancer cell death. Time-dependent proliferation assay showing reduced cell growth in (**A**) BxPC-3 and (**B**) PANC-1 cells following cell transfection with the respective plasmids. Fold changes are calculated with respect to pCMV Myc transfected cells. Data are presented as the mean ± SD of three experiments. *** represents *p* ≤ 0.001, and **** *p* ≤ 0.0001. (**C**) BxPC-3 and (**D**) PANC-1 cells were transfected with TLR2 and TLR9 intrabodies to detect cell death using propidium iodide (PI) staining and quantitation. The percentage of PI-positive cells is presented as the mean ± SD of three individual staining experiments. ** represents *p* ≤ 0.01, and **** represents *p* ≤ 0.0001. (**E**) Ki67 immunostaining (green) and its quantitation in PANC-1 cells show a reduced proliferation at 72 h post-transfection. Scale bar = 10 μm. ** represents *p* ≤ 0.01. Fold changes are calculated with respect to pCMV Myc NCAM1 transfected cells. Data are presented as the mean ± SD of *n* = 3 staining experiments. F-Actin (red) was used to label the cell body.

**Figure 2 antibodies-13-00011-f002:**
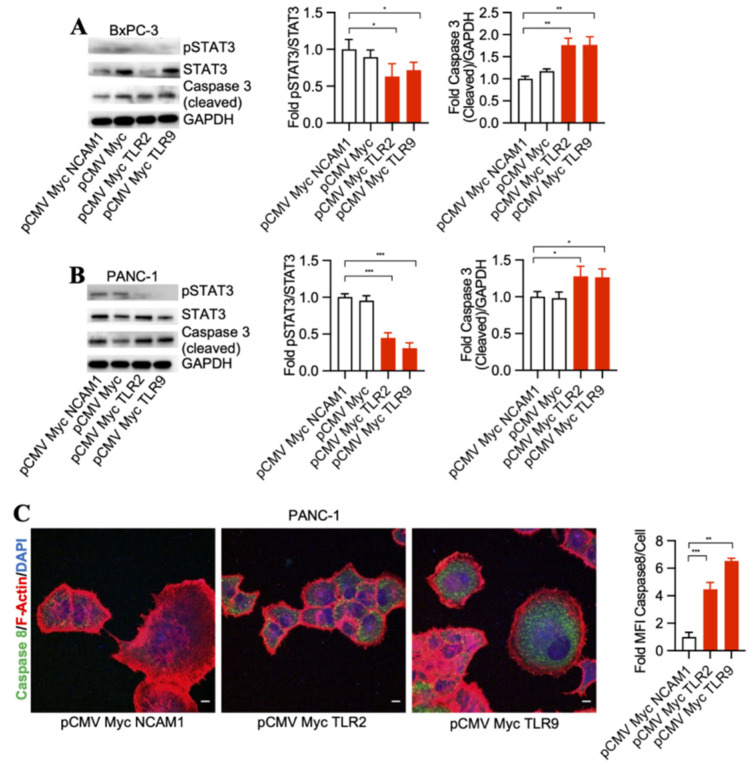
TLR2 and TLR9 intrabodies inhibit pancreatic cancer cell growth via the STAT3 signaling pathway. Western blot analysis in (**A**) BxPC-3 and (**B**) PANC-1 cells stably expressing TLR2 and TLR9 intrabodies demonstrated a decrease in STAT3 phosphorylation following the expression of TLR2 and TLR9 intrabodies and an increase of cleaved Caspase 3. GAPDH served as a loading control. Quantitation of pSTAT3 and cleaved Caspase 3 in both BxPC-3 and PANC-1 cells. (**C**) Increase in Caspase 8 expression in PANC-1 cells stably expressing TLR2 and TLR9 intrabodies. Fold changes are calculated with respect to pCMV Myc NCAM1 transfected cells. Data are presented as the mean ± SD of *n* = 3 experiments. F-Actin (red) was used to label the cell body. Scale bar = 10 μm. * represents ≤ 0.05, ** *p* ≤ 0.01, and *** represents *p* ≤ 0.001.

**Figure 3 antibodies-13-00011-f003:**
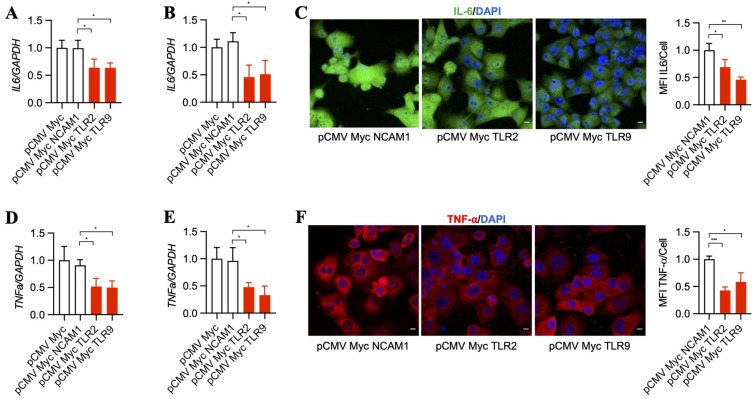
TLR2 and TLR9 intrabodies inhibit the production of inflammatory mediators IL-6 and TNF-α. TaqMan RT-PCR analysis for *IL-6* in (**A**) BxPC-3 and (**B**) PANC-1 cells stably expressing TLR2 and TLR9 intrabodies showed significantly decreased *IL-6* transcripts following TLR2 and TLR9 intrabody expression. Fold changes are calculated with respect to pCMV Myc transfected cells. Data are presented as the mean ± SD of three individual experiments. * represents *p* ≤ 0.05. (**C**) Immunostaining for IL-6 and its quantitation showed decreased expression following TLR2 and TLR9 intrabody administration and Brefeldin A treatment in PANC-1 cells stably expressing TLR2 and TLR9 intrabodies. Scale bar = 10 μm. Fold changes are calculated with respect to pCMV Myc NCAM1 transfected cells. Data are presented as the mean ± SD (n = 3). * represents *p* ≤ 0.05, and ** represents *p* ≤ 0.01. TaqMan RT-PCR analysis for *TNFa* in (**D**) BxPC-3 and (**E**) PANC-1 cells stably expressing TLR2 and TLR9 intrabodies showing significant decreased *TNFa* transcripts following TLR2 and TLR9 intrabody expression. Fold changes are calculated with respect to pCMV Myc transfected cells. Data are presented as the mean ± SD of n = 3 experiments. * represents *p* ≤ 0.05. (**F**) Immunostaining for TNF-α after Brefeldin A treatment and its quantitation showing decreased expression following TLR2 and TLR9 intrabody expression in PANC1 cells stably expressing TLR2 and TLR9 intrabodies. Scale bar = 10 μm. Fold changes are calculated with respect to pCMV Myc NCAM1 transfected cells. Data are presented as the mean ± SD. * represents *p* ≤ 0.05, and *** represents *p* ≤ 0.001; n = 3 experiments.

**Figure 4 antibodies-13-00011-f004:**
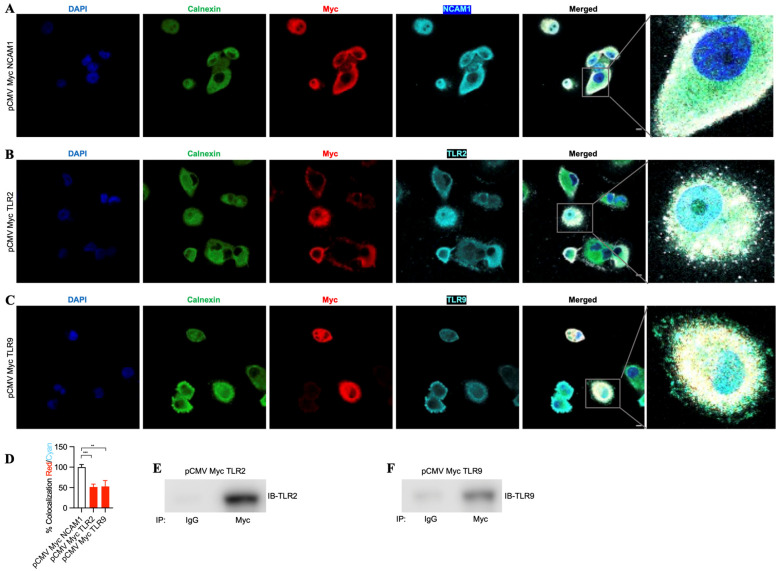
TLR-2- and TLR9-specific intrabodies bind to their target proteins and retain them in the endoplasmic reticulum. Co-immunostaining for Calnexin (green); IB-Myc (red); and target proteins (**A**) NCAM1, (**B**) TLR2, and (**C**) TLR9 (all cyan) show the colocalization of IB-Myc with its specific target, as well as colocalization with Calnexin in PANC-1 cells transiently transfected at 72 h post-transfection. Scale bar = 10 μm. Zoomed images for better visualization on the right panel. (**D**) Quantitation of the colocalization of IB Myc with NCAM1, TLR2, or TLR9, which was plotted as the percent colocalization after normalization to NCAM1 as 100%. ** represents *p* ≤ 0.01, and *** represents *p* ≤ 0.001; n = 3. Immunoprecipitation (IP) using anti-IgG (control) or anti-Myc antibodies and immunoblotting (IB) for (**E**) IB TLR2 and (**F**) IB TLR9 from immunoprecipitated samples show the specific binding of the respective intrabodies to their target protein.

**Table 1 antibodies-13-00011-t001:** List of the primers used for TaqMan PCR.

Gene Name	Assay ID
IL-6	Hs00174131_m1
TNFα1	Hs00855471_g1
GAPDH	Hs99999905_m1

## Data Availability

Data are contained within the article and Appendix A.

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
