# Peer review of "TLR2 and TLR9 Blockade Using Specific Intrabodies Inhibits Inflammation-Mediated Pancreatic Cancer Cell Growth"

_2073-4468, 2024, doi:10.3390/antib13010011_

Round 1

Reviewer 1 Report

Comments and Suggestions for Authors

Author Response

The authors describe the use of two ER-retaining intrabodies against TLR2 and 9 and their effect on PDAC cells proliferation, death and downstream pathways. This is an interesting study but important controls are missing. Hence, the authors should reply to the comments below:

L47: these numbers are not updated, this is now around 12%. Please use more recent references such as Siegel et al. Cancer statistics 2023 (DOI: 10.3322/caac.21763).

Response to the reviewer: We thank the reviewers for their comments and appreciate their efforts to improve our manuscript. We have added the new statistics and the reference (Ref. 1)

L50: mFOLFIRINOX is also an important chemotherapy used in PDAC. Overall, the introduction on pancreatic cancer should be revised with a more recent bibliography.

Response to the reviewer: We added a new reference (Ref. 3).

L91: a specific section within the introduction should be written on intrabody: what are tintrabodies? Advantages/over other strategies. Explain their functionalization within the ER retention with the KDEL motif. This is important for this manuscript to be accessible to a broad audience of scientists.  

Response to the reviewer: We have included a specific section on intrabodies and included the explanation for ER retention and added Ref. 30. Thank you.

L96-98: REF 25-28 are not recent as stated in this sentence (two of them are from 2005). “Recent” should be removed from this sentence or other references should be added. Also REF 28 is a review, research articles should be cited here.

Response to the reviewer: We have removed the word recent and the review

L129-141: by adding a section “Generation of intrabodies”, it is not clear whether the intracellular antibodies were selected in this study or from other studies (REFs 23,24). It seems that these come from previous studies, therefore the “Generation of intrabodies” section should be removed from the materials and methods section. The authors should rather explain the background of these antibodies in the introduction (see comment above on intrabodies).

Response to the reviewer: Thank you for your recommendation. We agree with the reviewer and removed the sentences.

L332-333: “we found that that TLR2 and TLR9 intrabodies resulted in significant cell death as compared to controls (pCMV Myc and pCMV Myc NCAM1) of BxPC-3 and PANC-1 cells”. This is overstatement as 1) it is really difficult to see something on these pictures (even in the supplementary figure), 2) there is no quantification (so significant should not be written) and 3) phase contrast is not a cell death marker per se.

Response to the reviewer: We removed the word significant and we amplified the supplementary figures. To our knowledge phase contrast microscopy has been used to observe the gross morphological changes associated with apoptosis such as cell shrinkage or membrane blebbing. It can even be used to differentiate between apoptosis and another type of cell death, necrosis.

Figure 1 is too small, it is very difficult to see something.

Response to the reviewer: Supplementary Figure 2 was amplified to complement Fig. 1.

L338-341: the authors should perform an Annexin V/PI staining rather than a simple PI staining to be accurate on their conclusions on the cell death.

Response to the reviewer: Fig. 1, Propidium iodide is commonly used to determine if cells are dead. It is a good indicator of cell viability, based on its capacity to exclude dye in living cells.  

Figure 2: The control of expression of the intrabodies is missing in Fig2A/B (using an anti-myc antibody for instance). Also, the authors should show as results the immunofluorescence of intrabodies (control, TLR2 and TLR9) in BxPC-3 and PANC-1 stable cell lines. There is a need of a proper characterization of the stable cell lines (make a supplementary figure for instance).

Response to the reviewer: We have shown the intracellular staining of TLR2 and 9 intrabodies by staining Myc in Figure 4. We used Myc antibody to detect TRL2 and 9 intrabodies. We performed western blotting to detect the Myc-tagged intrabody expression, but we were not able to detect the specific bands. Thus, we used immunostaining to detect the Myc-tagged intrabodies as they showed specific intracellular staining, which was verified by the endoplasmic reticulum marker.

The model used is not optimal: using stable cell lines constitutively expressing these intrabodies is not great as their transient expression in Fig1C inhibits PDAC cells proliferation. Therefore, an inducible system should be used rather than a constitutive system.

Response to the reviewer: The reviewer is absolutely right. But in 10 days we were not able to perform immunofluorescence of both stable cell lines with all controls.  The reviewer points to inducible expression of the intrabodies. Inducible promoters should be used if the intrabody will be tested in appropriate xenograft tumor mouse models. This will be the next step. In this manuscript we demonstrate significant inhibition of tumor growth using constitutive expression in pancreatic tumor cell lines to characterize the mode of action of the intrabodies.

The authors should discuss in the result section about the increase of cleaved caspase 3 observed in Fig2A/B (L361-370).

Response to the reviewer: we added this information about caspase 3 in the results and figure legends. Thank you.

The authors should display BxPC-3 Caspase 8 level as for PANC-1 cells or at least why they did not show the data on Fig2.

Response to the reviewer: We have used the staining for PANC-1 cells as proof of concept. We apologize but we did not stain the BxPC-3 cells as this was to show the proof of concept only.

Figure 3C/F: the authors should also show the data for BxPC-3 cells.

Response to the reviewer: We have used the staining for PANC-1 cells as proof of concept. We did not stain the BxPC-3 cells.

Figure 4A, B and C: L412/413: “We found significant co-localization of these three proteins indicating their binding (Figures 4A, B, and C).” This is an overstatement as no quantification is shown on Fig4A-C (IF images only). It is very difficult to see a co-localization on these pictures. Higher magnification should be provided (for instance a zoom in on 1/2 cell(s)). Why using transient transfection instead of the stable cell lines for the IF (and the IP experiment)? This should be explained in the manuscript.

The cyan colour is difficult to read on the Figure 4, it should be modified.

Response to the reviewer: We have performed calculations for the co-localization of Calnexin with Myc-tagged intrabodies. We have changed the color of the cyan to read it better. We have included the higher-magnification images in the revised figure. We found that transient expressions are higher than the stable cells. This is the reason we used transient expression in co-localization and IP experiments to get a better signal.

Figure 4E, F: a negative control is missing in these panels. The authors should also include their negative intrabody on each panel (E and F).

Response to the reviewer: In our experiments, we have used IgG as a negative control for binding specificity for TLR2 and TLR9 antibodies. As we think, NCAM-1 pull-down will not immunoprecipitate the TLR2 or TLR9 protein.

L455-458: the authors should increase the diversity of their references. Many self-citations and from another group only, other groups work on intrabodies.

Response to the reviewer: We have included more references from other groups in our revised manuscript.

L496-501: the authors should elaborate a bit more on the pancreatic tumor-specific delivery and cite appropriate research articles rather than a review that does not discuss this purpose.

Response to the reviewer: We have elaborated more on the pancreatic tumor-specific delivery and cited appropriate research articles. We have removed the review citations as well.

Reviewer 2 Report

Comments and Suggestions for Authors

The study explores the use of TLR2 and TLR9 blockade using specific intrabodies that are capable of targeting intracellular tumor associated antigens and neoantigens.  The study suggests that intrabody-mediated TLR inhibition in the ER could be a potential therapeutic intervention strategy for pancreatic cancer.  The conclusion is supported by comprehensive experiments and analysis.  The study found that TLR2 and TLR9 intrabodies significantly inhibited pancreatic tumor cell growth, and decreased IL-6 and TNF-α expression.

In general, this study is well-designed and well-structured, providing sufficient information.  However, there are a few areas that could be improved for clarity and completeness:

It would be beneficial to mention if any experiments were repeated to validate the results.  The number of samples in each group should be clearly stated in section 2.4.  While the statistical tests used are mentioned, it could be helpful to further detail the statistical methods, including what responses were compared between groups.

The paper could benefit from a more extensive discussion on the limitations of the study and the potential implications of the research.  For instance, it would be useful to know more about the potential side effects or risks associated with this therapeutic strategy, and whether the approach could be applied to other types of cancer.

Author Response

In general, this study is well-designed and well-structured, providing sufficient information.  However, there are a few areas that could be improved for clarity and completeness:

It would be beneficial to mention if any experiments were repeated to validate the results.  The number of samples in each group should be clearly stated in section 2.4.  While the statistical tests used are mentioned, it could be helpful to further detail the statistical methods, including what responses were compared between groups.
Response to the reviewer: We included the number of repeated experiments and number of samples in each group and added this information in section 2.4., and in the Figure legends.

The paper could benefit from a more extensive discussion on the limitations of the study and the potential implications of the research. For instance, it would be useful to know more about the potential side effects or risks associated with this therapeutic strategy, and whether the approach could be applied to other types of cancer.

Response to the reviewer: We discussed the potential side effects and risks associated with the therapeutic strategy and added some papers of other types of cancer that could be targeted.

Round 2

Reviewer 1 Report

Comments and Suggestions for Authors

The authors replied to several points raised before, which improved the overall quality of the manuscript. However, there are mistakes of figure citation within the manuscript (for instance figure 1). The authors should double check that.  

Author Response

We corrected the issues with the figures. Thank you.